# Effect of Delayed Refrigeration on the Microbial Carcass Contamination of Wild Boars (*Sus scrofa*)

**DOI:** 10.3390/ani11051434

**Published:** 2021-05-17

**Authors:** Beniamino Cenci-Goga, Alberto Amicabile, Musafiri Karama, Saeed El-Ashram, Cristina Saraiva, Juan García-Díez, Simone Finotti, Viviana Genna, Giampaolo Moretti, Riccardo Murari, Riccardo Muliari, Sabrina Bonizzato, Erica Lugoboni, Sabina Cassini, Caterina Dal-Ben, Luca Grispoldi

**Affiliations:** 1Dipartimento di Medicina Veterinaria, Università degli Studi di Perugia, 06126 Perugia, Italy; grisluca@outlook.it; 2Department of Paraclinical Sciences, Faculty of Veterinary Science, University of Pretoria, Onderstepoort 0110, South Africa; musafiri.karama@up.ac.za; 3Azienda Ulss 9 Scaligera—Via Valverde, 42-37122 Verona, Italy; alberto.amicabile@aulss9.veneto.it (A.A.); simone.finotti@aulss9.veneto.it (S.F.); viviana.genna@aulss9.veneto.it (V.G.); giampaolo.moretti@aulss9.veneto.it (G.M.); riccardo.murari@aulss9.veneto.it (R.M.); 4School of Life Science and Engineering, Foshan University, Foshan 528231, China; saeed_elashram@yahoo.com; 5Faculty of Science, Kafrelsheikh University, Kafr el-Sheikh 33516, Egypt; 6Department of Veterinary Sciences, School of Agrarian and Veterinary Sciences, University of Trás-os-Montes e Alto Douro, 5000-801 Vila Real, Portugal; crisarai@utad.pt; 7Veterinary and Animal Research Centre (CECAV), University of Trás-os-Montes e Alto Douro, 5001-801 Vila Real, Portugal; juangarciadiez@gmail.com; 8Istituto Zooprofilattico Sperimentale delle Venezie (IZSVe)—Sezione di Verona—Via S. Giacomo, 5-37135 Verona, Italy; rmuliari@izsvenezie.it (R.M.); sbonizzato@izsvenezie.it (S.B.); elugoboni@izsvenezie.it (E.L.); scassini@izsvenezie.it (S.C.); cdalben@izsvenezie.it (C.D.-B.)

**Keywords:** wild boar, game, meat, microbiological quality, refrigeration

## Abstract

**Simple Summary:**

The bacteria that contaminate meat after the death of the animal can come from multiple sources, such as the animal itself, the external environment and the operator who handles it. The prompt refrigeration of hunted game is generally considered an important step to prevent spoilage and meat alterations, although many logistic hindrances, such as animal weight and distance from the hunting area to the refrigerators, limit the meticulous adoption of the best procedures. We show that the bacterial population of wild boar carcasses is not correlated to the mere time from shot to refrigeration but is correlated to the refrigeration time from chilling to analysis. The results of our study revealed a correlation between the time from shot to analysis and from refrigeration to analysis but a lack of correlation between the time from shot to refrigeration.

**Abstract:**

The immediate refrigeration of meat after slaughter is a key issue for the proper storage and aging of meat. The industry standard cold chain relies on low temperatures and ventilation to lower the internal carcass temperature to 0–4 °C within the first 48 h, i.e., within four times the so-called semi-cooling time. On the other hand, for games, once bled and eviscerated, the carcass must be sent to a point where it can be sectioned or kept on air for maturation at refrigeration temperature. The precautions to observe are few and simple but essential: protect the meat and start the cooling process quickly. After preparing the animal (bleeding and evisceration), it may be necessary to face a period of transport that is sometimes long and not very easy; while small animals can be easily transported in a backpack, larger ones must necessarily be carried by several people or sometimes dragged to the vehicle capable of transporting them. It is obvious that a wild boar opened from the jaws to the pelvis and dragged for hundreds of meters will tend to be contaminated, although these contaminations are to be considered secondary for the preservation of the meat, compared to contamination by the intestinal contents. In an attempt to investigate the effect of delayed refrigeration on wild boar carcass contamination, the aim of this work was to determine a correlation between several hunting and logistic parameters (age, sex, animal weight, shooting distance, number of shots, weather and temperature and time from shot to refrigeration and to analysis) and bacterial contamination of the carcass. The correlation coefficient, r, was found to be 0.038 for the eviscerated body weight (*p* < 0.05), 0.091 for the external temperature on the day of hunting (*p* < 0.05), 0.027 for the time from shot to refrigeration (*p* = 0.081), 0.038 for the time from refrigeration to analysis (*p* < 0.05) and 0.043 for the time from shot to analysis (*p* < 0.05). These results stand for a negative correlation between the bacterial population and eviscerated carcass weight and between the bacterial population and external temperature and for a positive correlation between the time from shot to analysis and from refrigeration to analysis. No association was demonstrated between the bacterial population and the time from shot to refrigeration.

## 1. Introduction

Consumers are increasingly becoming concerned about healthy and safe products, and the demand for these products is escalating [1]. Consumers expect the meat products on the market to have the required nutritional values, to be wholesome; fresh; lean and have adequate juiciness, flavor and tenderness and be harvested with a high animal welfare standard [2,3,4]. Game and venison meat are, nowadays, perceived by some categories of consumers as more animal welfare-friendly, because hunted game meat is considered more ethically justifiable than farmed meat, since the animals roam free until the moment of harvest [5]. A recent study proved that German consumers strongly prefer hunting to intensive farming [6].

Boar hunting is mainly divided into two categories: (i) collective hunting, consisting of the presence of a large number of participants with drive hunts, and (ii) individual hunting, carried out by a single hunter or by a number not exceeding three by stalking or posting [7]. Both rifled-bore and smooth-bore weapons can be used for wild boar hunting. In addition to the metal sights, the weapon can be set with red dot or telescope attachments. For smooth-bore weapons, only 12- or 20-gauge dry bullet ammunitions are allowed. These guns can be both swinging (like shotguns) and fixed (like semi-automatic). Instead, for rifled-core weapons, there are various models such as: bolt-action, express, falling block, lever and semi-automatic. The recommended caliber of the latter is from 7 mm upwards. The weapon of choice for wild boar hunting depends on the type of hunt and the local regulations [8].

Once bled, eviscerated and cooled, the carcass must be sent to a point where it can be sectioned or kept refrigerated for maturation. The precautions to be observed during this phase are few and simple but essential: protect the meat and start the cooling process [9]. After preparing the animal (bleeding, evisceration and cooling), it may be necessary to face a period of transport that is sometimes long and not very easy; while small animals can be easily transported in a backpack, larger ones must necessarily be carried by several people or sometimes dragged to the vehicle capable of transporting them. It is obvious that a wild boar opened from the jaws to the pelvis and dragged for hundreds of meters will tend to be contaminated with blades of grass and, sometimes, earth; however, these contaminations are to be considered secondary in the preservation of the meat, compared to the contamination by the intestinal contents, and can be easily eliminated with a simple washing on arrival at home [9]. The practice of not opening the thorax of a prey before arriving home (the abdomen is still opened due to weight problems during transport) is instead absolutely wrong, as it hinders the cooling of the meat and can favor the exit of liquids from the esophagus, liquids capable of contaminating the meat due to their bacterial components. Of the two types of meat contamination, ground contamination is much more tolerable, but it is still preferable to avoid it; it will be possible to have a plasticized sheet to tie around the torso of the animal, which, in the case of not excessively uneven terrain, as well as protecting the carcass, will facilitate its sliding [10]. For our areas, among the animals that cause the greatest transport problems, due to their size wild boars certainly apply. As soon as you arrive at a motorized vehicle capable of transporting the carcass, it is necessary to remember that this is used to transport the prey as soon as possible to a suitable place to start the cooling process (cold room or cold cellar) and not to transport the hunters in a suitable place to celebrate the capture, forgetting the prey perhaps under the sun; it is, in fact, essential to be able to quickly put the meat away from possible alterations [9,10]. Once at home, the animal is hung, washed on the inside of the carcass with plenty of water to clean off the dirt and blood that will pour out during the transport without wetting the hair and leave it hanging to drain off; once drained and dried (be careful not to put water where it is unable to drain—for example, inside the wounds), the carcass will be ready to pass into maturation and subsequent sectioning. During the draining and drying phases, it would be good to keep the chest of the carcass open with a special retractor (wooden stick) in order to favor its internal drying and, also, keep it in place in the cell, until cooling has taken place. It is necessary to remember the role of the external temperature in the processing and conservation of meat; the storage temperatures of the meat are 3–7 °C, but the cells are generally set at −1 °C, as, at this temperature, the meat does not freeze, so, if we have external temperatures of 18–20 °C (in the summer or autumn) and a cool cellar at 8 °C, this will be suitable to favor cooling but not suitable for the preservation of the carcass, which, if stored in that place, will tend to rot rapidly.

Refrigeration is done in cold rooms, at temperatures ranging from −1 to 3 °C. The cold acts on the meat by slowing down both the degradation processes of its nutrients, proteins and fats, in particular, and the proliferation of germs and moulds, which, in a more-or-less abundant way, contaminate the carcass. It is clear that, to obtain prolonged conservation, the cooling must be carried out as quickly as possible after recovery of the carcass and after the latter has undergone the practices of bleeding and evisceration, scrupulously observing the hygiene rules. This is of the utmost importance, because the bacteria that contaminate the meat in these phases drastically reduces its shelf life, due to the fact that many bacteria have adapted to survive in the soil and surface waters of cold areas, and so, they replicate well, albeit more slowly, at refrigeration conditions [11]. Furthermore, low temperatures also slow down the activities of the enzymes responsible for maturation, which will require longer times for their development. In addition to the temperature, humidity and air speed must also be considered for the purposes of maturation inside the refrigerator. These factors, if well-regulated, contribute to keep the bacterial population low on the surface and, at the same time, favor the maturation of the meat in depth. Refrigeration therefore performs two functions at the same time: it prolongs the life of the meat by promoting its maturation. It is clear that this method preserves the meat for a limited period of time, which, for furred game, hardly exceeds 10–15 days (in larger animals). The bacteria that contaminate the meat after the death of the animal can come from multiple sources, such as the animal itself, the external environment and the operator who handles it [11]. Perishable foods such as meats can be easily polluted, so it is important to strictly observe the hygiene rules to avoid bringing in bacterial colonies capable of drastically reducing its storage time (spoilage bacteria, e.g., *Pseudomonas* spp.) or even make it dangerous for the health of the consumer (pathogenic bacteria, e.g., *Salmonella* [12,13,14]. The speed with which game meat, from the initial contamination, undergoes the alterative process of putrefaction depends to a decisive extent of the type of bacteria present and their quantity, as well as on the atmospheric temperature, which can influence their multiplication in different ways [11].

In an attempt to investigate the effects of delayed refrigeration on wild boar carcass contamination, the aim of this work was to determine a correlation between several hunting and logistic parameters (age, sex, animal weight, shooting distance, guns caliber, number of shots, weather and atmospheric temperature, time from shot to refrigeration and to analysis) to the bacterial population of the carcass.

## 2. Materials and Methods

### 2.1. Animals and Hunting Conditions

One hundred and twelve wild boars were hunted by collective hunting in Verona Province (Figure 1) from June 2020 to January 2021. The distribution of animals is shown in Table 1, which gives the frequency distribution of the animals by the different categories.

After the shot animals were dressed in place and transported to the refrigeration chamber within an average delay of 3.9 h (SD 4.83, minimum 0.0 h and maximum 25.25 h), sampling occurred after 107.28 h (SD 71.598, minimum 3.5 h and maximum 283.5 h). Table 2 and Table 3 and Figure 2 show the distribution of the external temperatures per month (Table 3) and per hunting trip (Figure 2).

### 2.2. Colony Count

Counting of the mesophilic microorganisms at 30 °C was carried out according to ISO 4833-1: 2013 Microbiology of the food chain—Horizontal method for the enumeration of microorganisms—Part 1: Colony count at 30 °C by the pour plate technique. Briefly, the analysis was based on the counts of the colonies grown in Petri dishes containing Plate Count Agar (PCA) medium and sown according to the inclusion seeding technique (germ agar) after aerobic incubation at 30 °C for 72 h. The starting sample, represented by a sponge passed over the internal surface of the thoracic cavity (100 cm^2^), was resuspended in 100 mL of tryptone solution (diluent containing tryptone 1 g/L and sodium chloride 8.5 g/L). The initial suspension and its further decimal dilutions were then sown, at a rate of 1 mL for each sowing, in the same amount of Petri dishes, which were then filled with 12–15 mL of PCA kept melted at a temperature of 44–47 °C. After solidification, the plates were incubated aerobically at 30 °C for 72 h. For the calculation of the colony-forming units, plates corresponding to 2 successive dilutions were considered, in which the plate of the first dilution contained a number of colonies between 10 and 300.

### 2.3. Statistical Methods

Statistical analyses were performed with the aid of StatView for Mac OS (SAS Institute, Inc., Cary, NC, USA). Regression analysis and one-way ANOVA (Analysis of Variance) were used as a statistical test to assess the differences in the means between the groups. In order to find a relationship between the bacterial population associated to each hunting and logistic parameter, a regression analysis was conducted. To identify the risk factors associated with carcass contaminations above 5-log cfu cm^−2^, a univariate analysis of the variables of interest was first conducted with binary logistic regression, followed by multiple logistic regression.

## 3. Results and Discussion

The average bacterial count of the carcass surfaces was 3.27-log cfu cm^−2^ (SD 1.43, minimum 0.6 and maximum 7.04). Table 4 brings to light a lot of data. The most important findings should be discussed.

The relationship between the bacterial population and some risk factors, determined by a regression analysis, is shown in Figure 3.

The correlation coefficient, r, was found to be 0.038 for the eviscerated body weight (*p* < 0.05) (Figure 3a), 0.091 for the external temperature on the day of hunting (*p* < 0.05) (Figure 3b), 0.038 for the time from refrigeration to analysis (*p* < 0.05) (Figure 3c), 0.027 for the time from shot to refrigeration (*p* = 0.0811) (Figure 3d) and 0.043 for the time from shot to analysis (*p* < 0.05) (Figure 3e). Table 5 shows the average values for the factors correlated to the bacterial population split by contamination levels above and below 5-log cfu cm^−2^

These results stand for a negative correlation between the bacterial population and eviscerated carcass weight and between the bacterial population and external temperature and for a positive correlation between the time from shot to analysis and from refrigeration to analysis. No association was demonstrated between the bacterial population and the time from shot to refrigeration.

The multivariate analysis (Table 6 identified “temperature” as a factor inversely related to carcass contamination above 5-log cfu cm^−2^: OR 0.905 (CI = 0.842 − 0.972, *p* = 0.0062) indicates a lower probability of carcass contamination above 5 log in animals hunted with higher external temperatures.

However, the differences between the odd ratios from multiple logistic regression and simple logistic regression indicate that there are some confounding effects of other factors on the relationship between the external temperature and microbial population of a carcass. In fact, the simple logistic regression (Table 7) also showed statistically significant differences (*p* < 0.05) for “refr/sampling” (OR = 1.009, *p* = 0.0227) and “shot/sampling” (OR = 1.008, *p* = 0.0271), where, apparently, the longer the time, the higher the probability of a microbial population above 5-log cfu cm^−2^.

Figure 4 sheds some light: the percentile distribution of log cfu cm^−2^ (Figure 4a) clearly shows that, from the 50th percentile, the bacterial population of the carcasses is higher in wild boars hunted in the cold season. Figure 4b–d, on the other hand, shows that the time from shot to refrigeration, from refrigeration to analysis and, in general, from shot to analysis are higher in the cold or mild weather at the time of hunting.

This last result is of the utmost importance, because it is well-known that hunters are concerned by the external temperatures, and for hunting trip done during the hot season or in the warmest part of the day, they tend to shorten the delay between the shot and the proper refrigeration of the carcass. On the other hand, in the winter and the coldest hours of the day, hunters tend to delay carcass refrigeration, because they rely on low temperatures [10]. The results of our study, however, demonstrated that proper and prompt refrigeration is of the utmost importance also in the winter and in cold weather, because the carcass needs an immediate refrigeration for the proper storage and aging of meat. The industry standard cold chain, in fact, relies on low temperatures and ventilation to lower the internal carcass temperature to 0–4 °C within the first 48 h, i.e., within four times the so-called semi-cooling time.

Our study also demonstrated that the bacterial population of wild boar carcasses is not correlated to the mere time from shot to refrigeration but is correlated to the refrigeration time from chilling to analysis. The study results revealed a correlation between the time from shot to analysis and from refrigeration to analysis but a lack of correlation between the time from shot to refrigeration. This is another important point to consider, because the prolonged refrigerated storage of hunted meat, especially when the carcass has been contaminated by environmental microorganisms, may increase the probability of a higher bacterial population at the time of analysis. Our results demonstrated, however, a lower bacterial population in the heavier carcasses (Figure 3a), not confirmed by a univariate analysis. Furthermore, when the regression plot was rerun after omitting all the animals hunted in the winter or in the colder parts of the day, the time between the shot and refrigeration were positively correlated (Figure 5a): correlation coefficient, r, was 0.073, *p* < 0.05. On the other hand, the same test, run with only the animals hunted in the winter or in the coldest parts of the day (Figure 5b), showed no statistically significant differences (r = 0.006, *p* = 0.6112).

Data on the factors affecting the bacterial population in wild boar carcasses are available in the literature, although the results are contradictory. For instance, Avagnina, et al. [15] found that bacterial contamination—in particular, the aerobic viable counts—were not influenced by the time elapsed between shooting and evisceration and sampling. For these authors, the statistical analyses revealed that the aerobic viable counts and Enterobacteriaceae counts in each species did not differ significantly across either the shooting–evisceration or shooting–sampling time ranges. These authors postulated that a lack of any statistical significance could be related to the grouping of the data into time categories that subsequently reduced the sample sizes of each group. On the other hand, Stella et al. [11] found that the total viable counts were positively influenced by high environmental temperatures, and higher Enterobacteriaceae counts were detected in heavy male carcasses than females. More recently, Orsoni et al. [16] found that the aerobic colony count mean value in the whole sample was 4.67-log cfu cm^−2^ and varied with the boar’s total weight, time between evisceration and skinning and by cleaning. These authors found also that the aerobic colony counts increased with the boar’s total weight. Since a boar’s weight increases with age, they suggested that older animals may be more contaminated than younger ones and that their carcasses should therefore be managed with particular care. Our data regarding body weight proved that the contamination is higher for lighter carcasses during the regression analysis, but no association was statistically significant during the logistic regression (Figure 3a and Table 6). An experimental study by Soriano et al. [17] in wild red deer proved that no differences in the aerobic mesophilic bacteria of M. *Longissinumus dorsi* were detected for the carcasses eviscerated immediately or after 4 h.

The data on the average carcass contamination of hunted wild boar is very consistent across the literature and is distributed in the range of 3–6-log cfu cm^−2^ [9,11,15,16,17,18]. To assess the microbiological status of the wild boar carcasses analyzed, our data were compared with the criteria specified by Regulation (EC) No. 2073/2005. However, the reference to these criteria is for guidance only, since they are provided for the carcasses of domestic pigs slaughtered in licensed premises (slaughterhouses). According to EU microbiological criteria for pig carcasses, the results are interpreted as satisfactory if the daily mean log for the total viable counts is <4-log cfu cm^−2^, acceptable if between 4 and 5-log cfu/cm^−2^ and unsatisfactory if the daily mean log is >5-log cfu cm^−2^. In our study, the prevalence of total viable counts below 4-log cfu cm^−2^ (satisfactory) occurred in 82 out of 112 carcasses (73.2%), between 4 and 5-log cfu cm^−2^ (acceptable) occurred in 14 carcasses (12.5%) and above 5-log cfu cm^−2^ (unsatisfactory) occurred in 16 out of 112 (14.3%) carcasses (Table 1). The comparison of our results with the EU microbiological criteria that are used to demonstrate the microbiological quality of the production process indicates that the hygienic quality of the handling and dressing procedures with wild boar carcasses in our study was acceptable or satisfactory in 96 out of 112 carcasses (85.7%).

## 4. Conclusions

Our study showed that, in cold weather, hunters tend to delay the refrigeration of the carcass, and this delay is a clear confounder for the correlation between the bacterial population of the carcass and the time from shot to proper refrigeration. Another important result is, indeed, the time from refrigeration to sampling, since, for game, prolonged refrigeration is less effective if not applied promptly. From these data, we can draw three important conclusions: (i) timely refrigeration if a key parameter to limit the bacterial population on a carcass surface, (ii) delayed refrigeration can also have detrimental effects on the bacterial population of the carcass when the external temperature is low or extremely low and (iii) sampling for an analysis should not be delayed, because the proliferation of microorganism during refrigeration conditions occurs and may act as a confounder.

## Figures and Tables

**Figure 1 animals-11-01434-f001:**
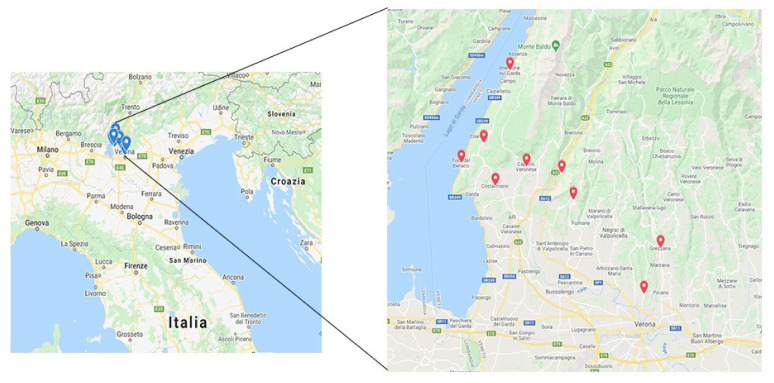
Map of the hunting sites.

**Figure 2 animals-11-01434-f002:**
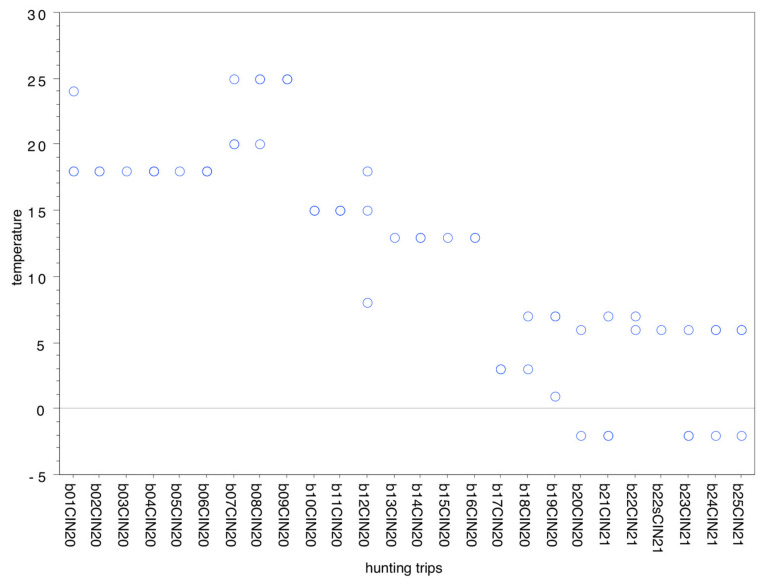
Atmospheric temperature in °C per hunting trip.

**Figure 3 animals-11-01434-f003:**
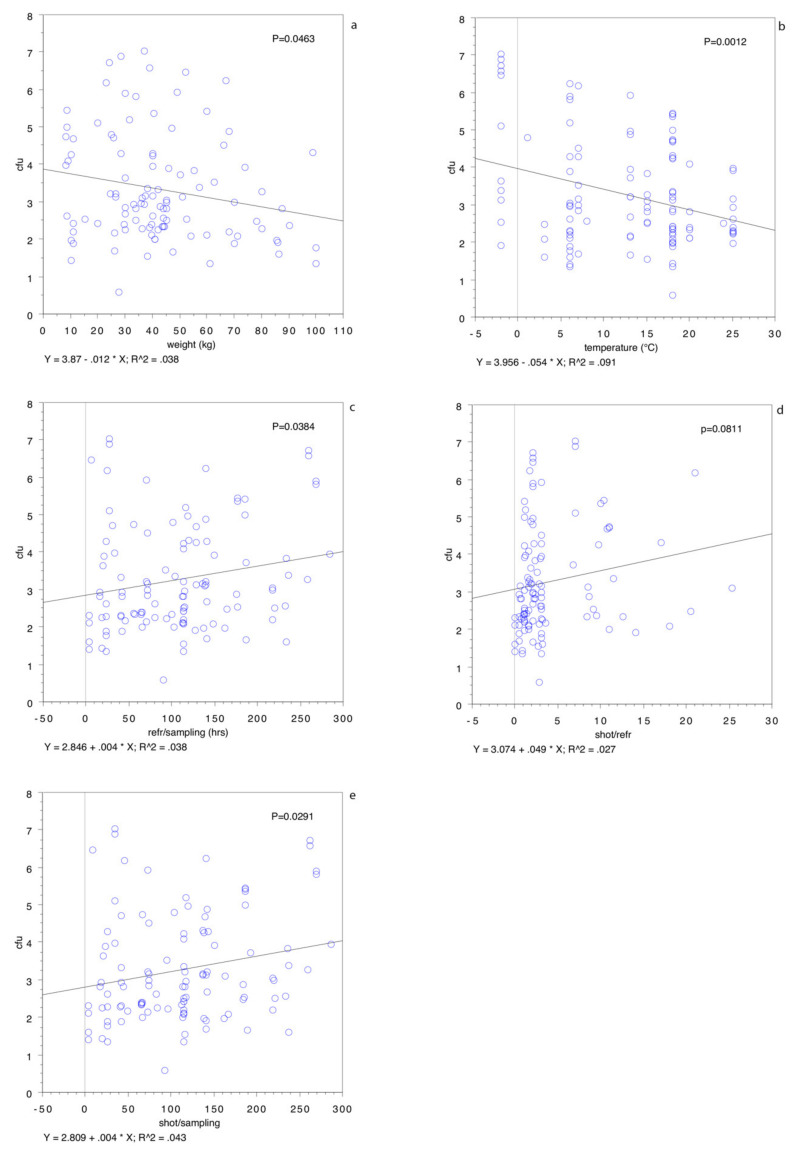
Regression plot for log cfu cm^−2^ and eviscerated carcass weight in kg (**a**), atmospheric temperature in °C (**b**), time from refrigeration to sampling in h (**c**), time from shot to refrigeration in h (**d**) and time from shot to sampling in h (**e**). cfu: log cfu cm^−2^.

**Figure 4 animals-11-01434-f004:**
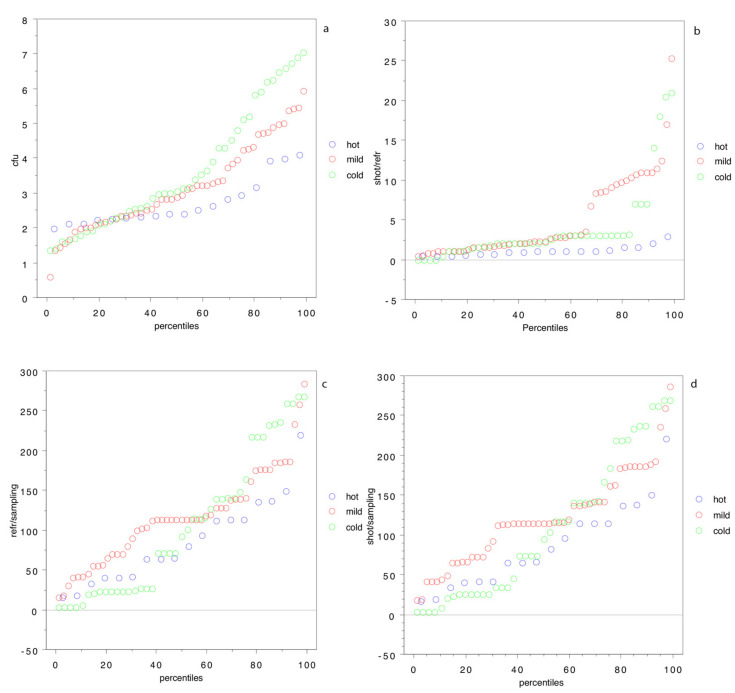
Percentile distribution by atmospheric temperature (hot >20 °C, mild 10–20 °C and cold <10 °C) for bacterial contamination of the carcass and, in log cfu cm^−2^, (**a**) for time from shot to refrigeration in h (**b**), time from refrigeration to sampling in h (**c**) and time from shot to sampling in h (**d**).

**Figure 5 animals-11-01434-f005:**
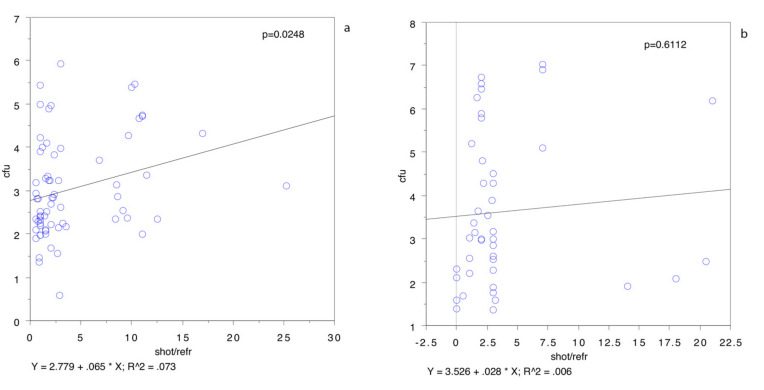
Regression plot for log cfu cm^−2^ and time from shot to refrigeration in h in carcasses from animals hunted in mild or hot atmospheric conditions, >10 °C, (**a**) and for animals hunted in cold atmospheric conditions, <10 °C (**b**).

**Table 1 animals-11-01434-t001:** Frequency distribution of samples divided by categories according to log cfu cm^−2^ values.

		Gender		Age					Weight		Distance		Shots		Weather				Temperature			Shot/Refr		Refr/Sampling		Shot/Sampling	
cfu	Total	m	f	Adult	Young	Red	Striped	Barn	<30 kg	>30 kg	<30 m	>30 m	>1	1	Cloudy	Serene	Rain	Variable	Hot	Mild	Cold	<3 h	>3 h	<48 h	>48 h	<48 h	>48 h
>0; <1	1	1	0	0	1	0	0	0	1	0	0	1	1	0	0	1	0	0	0	1	0	1	0	0	1	0	1
>1; <2	15	8	4	9	1	1	2	2	4	11	6	9	10	3	4	10	0	0	1	6	8	10	5	7	8	7	8
>2; <3	44	25	16	30	6	5	0	2	7	37	10	34	38	5	4	34	1	0	13	20	11	30	14	12	32	11	33
>3; <4	22	12	9	16	3	3	0	0	3	19	3	19	19	3	1	14	3	1	2	11	9	15	7	3	19	3	19
>4; <5	14	7	7	5	2	4	3	0	8	6	6	8	11	2	2	11	1	0	2	8	4	7	7	3	11	3	11
>5; <6	9	6	3	3	1	3	2	0	3	6	4	5	7	2	0	9	0	0	0	5	4	5	4	1	8	1	8
>6; <7	6	4	2	3	2	1	0	0	3	3	3	3	6	0	3	3	0	0	0	0	6	4	2	3	3	3	3
>7; <8	1	1	0	1	0	0	0	0	0	1	1	0	1	0	0	1	0	0	0	0	1	0	1	1	0	1	0
Total*	112	64	41	67	16	17	7	4	29	83	33	79	93	15	14	83	5	1	18	51	43	72	40	30	82	29	83

cfu: log cfu cm^−2^; gender: male (m) or female (f); age: adult (>22 months), young (12–22 months), red: 4–12 months), striped (0–4 months), barn (bred in a farm); weight: eviscerated carcass weight in kilograms >30 kg or <30 kg; distance: shooting distance in meters >30 m or <30 m; shots: number of shots (>1 or just 1); weather: cloudy, serene, rainy; variable; temperature: hot (>20 °C), mild (10–20 °C), cold (<10 °C); shot/refr: time in h from shot to refrigeration; refrl/sampling: time in h from refrigeration to sampling; shot/sampling: time in h from shot to sampling.*: some data is missing therefore totals do not always match the number of animals harvested.

**Table 2 animals-11-01434-t002:** Time in hrs from shot to refrigeration, from refrigeration to sampling and from shot to sampling distributed by atmospheric temperature.

	Shot/Refr				Refr/Sampling				Shot/Sampling			
	Total	Hot	Mild	Cold	Total	Hot	Mild	Cold	Total	Hot	Mild	Cold
mean	3.91	1.09	4.96	3.85	107.28	85.28	116.73	105.29	111.19	86.37	121.68	109.14
std. dev.	4.83	0.61	5.12	5.04	71.60	53.86	58.63	89.23	71.88	53.92	58.92	89.12
std. err.	0.46	0.14	0.72	0.77	6.77	12.70	8.21	13.61	6.80	12.71	8.25	13.60
n	112	18	51	43	112	18	51	43	112	18	51	43
minimum	0.00	0.50	0.50	0.00	3.50	16.02	16.02	3.50	3.50	16.75	18.33	3.50
maximum	25.25	2.92	25.25	21.00	283.50	220.00	283.50	267.50	286.50	221.00	286.50	269.50

hot: external temperature >20 °C, mild: temperature 10–20 °C, cold temperature <10 °C; shot/refr: time in h from shot to chiller; refr/sampling: time in h from chiller to sampling; time in h from shot to sampling, shot/sampling: time in h from shot to sampling.

**Table 3 animals-11-01434-t003:** Average temperature in °C of hunting trips divided per months.

	Mean	std. dev.	Minimum	Maximum	n. of Trips
total	12.69	7.92	−2.00	25.00	112
Jun-2020	24.00		24.00	24.00	1
Jul-2020	18.33	1.29	18.00	25.00	33
Aug-2020	22.06	3.98	15.00	25.00	17
Sep-2020	14.00	2.65	8.00	15.00	7
Oct-2020	18.00		18.00	18.00	1
Nov-2020	11.46	3.76	3.00	13.00	13
Dec-2020	3.67	3.81	-2.00	7.00	9
Jan-2021	4.03	3.72	-2.00	7.00	30
Feb-2021	−2.00		−2.00	−2.00	1

**Table 4 animals-11-01434-t004:** Average values for carcass contamination in log cfu cm^-2^ for the various groups of animals.

		Age								Gender			Weight	
	Total	Adult	Yearlings		Red			Striped	Barn	Male		Female	<30 kg	>30 kg
mean	3.27	3.11	3.44		3.88			3.84	1.86	3.34		3.37	3.62	3.14
std. dev.	1.43	1.34	1.65		1.45			1.53	0.43	1.53		1.29	1.64	1.34
minimum	0.60	1.36	0.60		1.90			1.45	1.40	0.60		1.56	0.60	1.36
maximum	7.04	7.04	6.91		6.74			5.46	2.32	7.04		6.58	6.91	7.04
	distance				shots									
	total	<30 m		>30 m	>1		1							
mean	3.27	3.61		3.12	3.32		3.12							
std. dev.	1.43	1.77		1.24	1.43		1.52							
minimum	0.60	1.36		0.60	0.60		1.36							
maximum	7.04	7.04		6.74	7.04		5.90							
	weather								temperature					
	total	cloudy		serene	rain	variable			hot	mild		cold		
mean	3.27	3.46		3.29	3.44	3.28			2.70	3.16		3.63		
std. dev.	1.43	1.92		1.45	0.56	.			0.68	1.25		1.75		
minimum	0.60	1.36		0.60	2.82	3.28			1.98	0.60		1.36		
maximum	7.04	6.74		7.04	4.32	3.28			4.11	5.93		7.04		
	shot/refr				refre/sampling				shot/sampling					
	total	<3 h		>3 h	<48 h	>48 h			<48 h		>48 h			
mean	3.27	3.14		3.49	3.23	3.28			3.27		3.27			
std. dev.	1.43	1.38		1.51	1.68	1.34			1.69		1.34			
minimum	0.60	0.60		1.36	1.36	0.60			1.36		0.60			
maximum	7.04	6.74		7.04	7.04	6.74			7.04		6.74			

age: adult (>22 months), yearlings (12–22 months), piglets red: (4–12 months), piglets striped (0–4 months), barn (bred in a farm); gender: male (m) or female (f); weight: eviscerated carcass weight in kilograms <30 kg or >30 kg; distance: shooting distance in meters <30 m or >30 m; shots: number of shots (>1 or just 1)weather: cloudy, serene, rainy; variable; temperature: hot (>20 °C), mild (10–20 °C), cold (<10 °C); shot/refr: <3 h or >3 h from shot to refrigeration; refr/sampling: <48 h or >48 h from refrigeration to sampling; shot/sampling: <48 h or >48 h from shot to sampling.

**Table 5 animals-11-01434-t005:** Average values for eviscerated carcass weight, external temperature, refrigeration to sampling time and shot to sampling time for the two groups with log cfu cm^−2^ <5 or >5.

	Weight (kg)			Temperature (°C)			Refr/Sampling (h)			Shot/Sampling (h)		
	Total	<5	>5	Total	<5	>5	Total	<5	>5	Total	<5	>5
mean	42.55	42.40	43.43	12.69	13.56	7.44	107.28	100.79	146.22	111.19	104.89	149.02
std. dev.	22.53	23.39	17.57	7.92	7.68	7.52	71.60	70.48	67.71	71.88	70.92	67.83
count	105	89	16	112	96	16	112	96	16	112	96	16

weight: eviscerated carcass weight in kilograms; temperature: in °C; refr/sampling: time in h from refrigeration to sampling; shot/sampling: time in h from shot to sampling, <5 or >5: average value for carcasses with log cfu cm^−2^ <5 or > 5.

**Table 6 animals-11-01434-t006:** Factors associated with carcass contamination >5 log cfu cm^−2^: results of multiple logistic regression.

	OR (95% C.I.)	*p*
weight (kg)	0.994 (0.966—1.023)	0.5087
temperature (°C)	0.899 (0.833—0.970)	*p* < 0.05 *
cell/sampling (hrs)	1.117 (0.935—1.334)	0.2220
shot/sampling (hrs)	0.900 (0.833—0.970)	0.2496

OR: odd ratio; *P*: *P*-value; C.I.: confidence interval; *: *p* < 0.05.

**Table 7 animals-11-01434-t007:** Factors associated with carcass contamination >5 log cfu cm^-2^: results of logistic regression for each variable.

	OR (95% C.I.)	*p*
weight (kg)	1.002 (0.979–1.026)	0.865
temperature (°C)	0.905 (0.842–0.972)	*p* < 0.05 *
cell/sampling (hrs)	1.009 (1.001–1.016)	*p* < 0.05 *
shot/sampling (hrs)	1.008 (1.001–1.016)	*p* < 0.05 *

OR: odd ratio; P: P-value; C.I.: confidence interval; *: *p* < 0.05.

## Data Availability

Not applicable.

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
