# Peer review of "Effect of Delayed Refrigeration on the Microbial Carcass Contamination of Wild Boars (Sus scrofa)"

_animals, 2021, doi:10.3390/ani11051434_

Round 1

Reviewer 1 Report

Dear Authors, congratulation for the work. The topic is very interesting and the text well written. My suggestion to improve the quality of the manuscript is to double check minor errors in spelling. 

Author Response

 Dear Professor Phillips,

please find enclosed the answers to reviewers for Manuscript ID animals-1192309« Effect of delayed refrigeration on the microbial population of hunted wild boars».

We apologize with reviewers, but the system has automatically reformatted the tables making them impossible to read.

Please find enclosed below a table with the list of reviewers’ comments and suggestions and our replies to the right.

Yours sincerely,

Beniamino Cenci-Goga

Reviewer 2 Report

Lines 44 – The authors are talking about bacterial populations, and not “load”. I would prefer that they correct the terminology throughout the manuscript.

Lines 149 – 150 The figure 1 provided to me is a representation of atmospheric temperature by hunting trip. While this is useful information, it does not seem to match the text in lines 149-150.

Line 150 and Table 1      The table requires a little more explanation. The number of shots do not seem to match the number of animals harvested. For a cfu of 1-2, there were 15 animals harvested, but only 13 are accounted for under the number of shots column.  This is true for the 2-3 and 4-5 rows as well.

Line 156 – Populations in place of “counts”, please.

Line 178 – Is this the average population? I would assume that it is.

Lines 192 – 194 The authors need to review this sentence for accuracy. It appears to be contradictory as written.

Lines 197 – 200  This certainly makes sense from a biological perspective. The microbial populations will increase given a longer incubation period.

General Comment: I found this an interesting manuscript to read, and it should provide useful information to those who hunt wild game.

Author Response

(The authors gave the same response as above.)

Reviewer 3 Report

The manuscript Effect of Delayed Refrigeration on the Microbial Load of Hunted Wild Boars (Sus scrofa) in Northern Italy presents the influence if time from the shot to refrigeration, ambient temperatures, age and sex on the bacterial contamination of hunt-harvested wild boars.

General comments

The manuscript has to be rejected, as in the current form it’s not suitable for publication.

  • The introduction is too long – see detailed comments.
  • The statistical analysis are incorrect, and there is a general chaos in statistics. In the statistical analysis chapter, the Authors claim that the one-way ANOVA was used, but then in the text (line…) a multifactorial ANOVA is mentioned?
  • Tables:

Tables are very confusing, it is hard to define what they present.

If you have used the multifactorial ANOVA, then the effect of all the factors (temperature, time from A to B procedure, age, sex etc) on the cfu load should be presented in tables. This gives huge tables, and the p-values for all the effects and interactions between these effects should be also considered. It would be more simple to use one-way ANOVA and consider each effect separately.

  • Figures – they are not described, and should be. Maybe correlation coefficients should be placed in a table instead of so many figures?

For details, see detailed comments

  • Results and discussion

The results of this experiment are discussed in general, but not with the available literature. This means, that a huge part of the paper is lacking.

Detailed comments

Title ‘Effect of Delayed Refrigeration on the Microbial Load of 2 Hunted Wild Boars (Sus scrofa) in Northern Italy’ makes the application of these research results very narrow. You can give the data on location in the Material and methods section, but by placing it in the title you diminish the value of you work.

Introduction

The lines 52-72 are too general, I would delete this part. This will make the introduction shorter, and will limit the cited literature to most valuable papers.

Table 1. - The table in unacceptable. It is absolutely unclear. Maybe the Authors could divide it to a few tables? Besides, as mentioned before, this kind of table is a results of multifactorial ANOVA, which means that you should give p-values for all the analysed effects separately and for all the interactions. And – I do not understand what Table 1 really presents?

Table 2. – shot/ cell – I do not understand what it means, however I assume that maybe it’s connected with the guns calibre, number of shots? But it’s not stated. The only data presented should be the mean value, SD or SE, p-value. But why these temperatures appear? The improper look of the table is most probably a result of improper ANOVA model.

Table 4. ‘red, stripe, barn’ – I don’t understand what was analysed exactly? Once again the whole table requires conversion. I don’t see the need for presenting min and max values. Use standard error instead, it will be more valuable.

Table 6. Please explain C.I. abbreviation

P values are normally given up to 3rd decimal place, and if the p values are given in tables, there's no need to repeat the significance with *. So use either exact p-values or P<0.05 marked with *

Author Response

(The authors gave the same response as above.)

Round 2

Reviewer 3 Report

Dear Authors.

The manuscript has significantly improved, all my concerns have been addressed. The statistical analysis looks good, and the tables are also proper.

There still are a few issues that require attention - I've used the review mode and made the necessary corrections/suggestions directly on the pdf file of the manuscript.

Best regards

Author Response

We have included all suggestions and all authors which the express gratitude for your time and professional advice.

Beniamino Cenci Goga
